# Molecular Characteristics of Cell Pyroptosis and Its Inhibitors: A Review of Activation, Regulation, and Inhibitors

**DOI:** 10.3390/ijms232416115

**Published:** 2022-12-17

**Authors:** Shaoqiang Wei, Min Feng, Shidong Zhang

**Affiliations:** Engineering Technology Research Center of Traditional Chinese Veterinary Medicine of Gansu Province, Lanzhou Institute of Animal Husbandry and Pharmaceutical Sciences, Chinese Academy of Agricultural Sciences, Lanzhou 730050, China

**Keywords:** pyroptosis, inflammatory caspase, inflammasome, pyroptosis blockers, herbal medicine

## Abstract

Pyroptosis is an active and ordered form of programmed cell death. The signaling pathways of pyroptosis are mainly divided into canonical pathways mediated by caspase-1 and noncanonical pathways mediated by caspase-11. Cell pyroptosis is characterized by the activation of inflammatory caspases (mainly caspase-1, 4, 5, 11) and cleavage of various members of the Gasdermin family to form membrane perforation components, leading to cell membrane rupture, inflammatory mediators release, and cell death. Moderate pyroptosis is an innate immune response that fights against infection and plays an important role in the occurrence and development of the normal function of the immune system. However, excessive pyroptosis occurs and leads to immune disorders in many pathological conditions. Based on canonical pathways, research on pyroptosis regulation has demonstrated several pyroptotic inhibitors, including small-molecule drugs, natural products, and formulations of traditional Chinese medicines. In this paper, we review the characteristics and molecular mechanisms of pyroptosis, summarize inhibitors of pyroptosis, and propound that herbal medicines should be a focus on the research and development for pyroptosis blockers.

## 1. Introduction

The modes of cell death include programmed and non-programmed cell death. Programmed cell death refers to the programmed process of cell death in order to maintain the stability of the internal environment after cells receive a certain signal or are stimulated by certain factors. Apoptosis, autophagy, programmed necrosis, and pyroptosis are the manifestations of programmed death. Thereinto, pyroptosis is a newly discovered form of programmed cell death, which was firstly discovered by Zychlinsky et al. in macrophages infected with *Shigella flexneri,* because its morphological characteristics were significantly different from apoptosis [1]. Later, Cookson et al. found that it is dependent on the activity of caspase-1 and different from caspase-3-activity-dependent apoptosis, defined this form of cell death as caspase-1-dependent cell death for the first time, and proposed the concept of pyroptosis for the first time [2]. Due to the dependency on inflammatory caspase-1, pyroptosis primarily refers to inflammatory cell death, which is obviously different from apoptosis and necrosis. Meanwhile, with the deepening of research on pyroptosis, it is found that pyroptosis plays an important role in the occurrence and development of many diseases. 

Pyroptosis is an important innate immune response that is critical in fighting infection. It is characterized by the expansion of cells until the cell membrane ruptures, resulting in the release of intracellular contents and activating the body’s strong inflammatory response. IL-1β and IL-18 released by pyroptosis cells are endogenous immune factors that cause fever, stimulate the activation of immune cells, and promote lymphocyte proliferation and the secretion of antibodies, while excessive pyroptosis leads to the uncontrolled release of IL-1β and IL-18, which will lead to a wide range of inflammatory reactions and immune diseases [3]. 

Pyroptosis is a double-edged sword; moderate pyroptosis contributes to the stability of the intracellular environment and plays an important role in fighting infection through eliminating bacteria to protect the host, but excessive pyroptosis not only leads to immune desensitization, but also causes life-threatening diseases such as sepsis, cytokine release syndrome (CRS), severe inflammation, and tissue damage [4,5,6,7]. Thus, excessive pyroptosis warrants serious attention during disease treatment. In order to decrease the adverse effects of excessive pyroptosis, the potential pyroptotic blockers have been extensively explored recently, which can keep pyroptosis within a reasonable range through inhibiting or regulating the pathway of pyroptosis. Understanding the occurrence and regulatory mechanism of pyroptosis and identifying potential inhibitory drugs may provide a new direction for further research on pyroptosis. 

## 2. Characteristics of Pyroptosis

### 2.1. Morphological Characteristics of Pyroptosis

Pyroptosis has some characteristics of necrosis and apoptosis in morphology, but it is different from apoptosis and necrosis. When cells undergo pyroptosis, cytoplasm membrane is ruptured and forms between 1–2 nm in diameter holes, resulting in the release of intracellular substances, the outflow of potassium ions, and cell swelling [8]. The substances stimulate the body’s immune response, recruits more inflammatory cells, and expands the inflammatory response. In this process, the nuclei gradually becomes round and nuclear condensation occurs, and the chromatin DNA breaks and degrades randomly [9,10,11]. 

### 2.2. Molecular Characteristics of Pyroptosis

#### 2.2.1. Inflammatory Caspases

Caspases were first discovered from nematodes in 1993 and are mainly involved in cell apoptosis and inflammatory reaction [12]. At present, 15 caspase family members have been found in mammals, including 13 caspases in humans and 11 caspases in mice [13]. Caspases are evolutionally conserved intracellular proteases with homology and similar structural features. Their active sites contain cysteine residues that can specifically cleave the peptide bond behind the aspartic acid residues of target proteins and are known as aspartic acid-specific cysteine proteolytic enzymes [14]. Caspase has been proved to be an essential protease for the developmental death of biological somatic cells. In normal cells, caspase usually exists in an inactive proenzyme state (pro-caspase) that can become active caspases after hydrolysis of amino acid sequences, thereby cutting relevant substrates, leading to the activation, inactivation, repositioning, or remodeling of substrates to play its role [15]. 

According to the differences in structure and function, caspases can be divided into apoptotic and inflammatory classes. Apoptotic caspases were related to apoptosis including caspase-2/3/6/7/8/9/10 and represented by caspase-3, while inflammatory caspases mediate inflammatory reaction and pyroptosis including caspase-1/4/5/11/12/13/14 [16,17]. Among inflammatory caspases, mouse caspase-1/11 or human caspase-1/4/5 are key proteins mediating pyroptosis pathway. Caspase-1, the first identified member of the caspase family, is responsible for cleaving pro-interleukin-1beta (pro-IL-1β). In addition, caspase-1 can cleave pro-interleukin-18 (pro-IL-18) into the mature form of IL-18 to play an immunomodulatory function [18,19,20,21,22,23]. Caspase-11 exists only in rodents, and mouse caspase-11 and human caspase-4/-5 are evolutionarily homologous genes [3]. Functionally, both human caspase-4/-5 and mouse caspase-11 can recognize LPS in cells ·. 

In addition to executioner caspases, activation of almost all caspases requires removal of the interdomain linker (IDL) and the prodomain [24,25,26,27,28,29]. The prodomain has multiple functions: first, the removal of the prodomain is an important step in caspase activation; second, the prodomain plays an important role in promoting caspase; third, the predomain has a stabilizing effect, and removal of the predomain would inactivate the caspase [21,26,30,31,32,33,34,35,36,37,38]. As shown in Figure 1, pro-caspase-1 also shares these common sequence features, with its N-terminal prodomain consisting of CARD (residues 1–95), a CARD domain adaptor (CDL) to the caspase domain, and an IDL between the p20 and p10 subunits. The catalytic C285 cysteine residue is located within the P20 region; for instance, self-proteolytic cleavage occurs at three aspartate residues (D103, D119, D297, and D316), releasing the prodomain (CARD, CDL, IDL), which is a key step in human caspase-1 activation [21,39]. There are two models for caspase-1 activation. The first model proposes that pro-caspase-1 dimerization is followed by self-proteolytic cleavage, while the second model has self-proteolytic cleavage before dimerization [40]. Recent results are more consistent with the previous dimerization–autoproteolytic model [32,41]. Recruitment of pro-caspase-1 to the inflammasome platform contributes to its activation. Various reports have indicated that caspases are activated by substrate-induced or adjacent-induced oligomerization followed by autoproteolytic cleavage [21,28,29,31,32,41,42,43,44,45,46,47]. 

Caspase-11 is essentially an endogenous receptor of LPS. For caspase-11 activation, LPS directly binds to the CARD domain of caspase-11 via its lipid A tail [48]. In the case of Gram-negative bacterial infections, the hexacylated lipid A moieties of LPS bind directly to the caspase-11 CARD domain, resulting in oligomerization and adjacent induced activation of caspase-11, which may be facilitated due to the polymeric tendencies of LPS. Cytoplasmic pentacylated and hexacylated LPS could induce caspase-11 activation. However, this was not observed for tetracylated LPS [49]. The main effects of caspase-11 are associated with non-canonical pyroptosis [50]. During non-canonical pyroptosis, caspase-11 directly cleaves GSDMD as non-canonical inflammasome [51].

#### 2.2.2. Gasdermin Family

Gasdermin (GSDM) is a family of pore-forming effector proteins discovered in recent years, and the GSDM family consists of Gasdermin A (GSDMA), Gasdermin B (GSDMB), Gasdermin C (GSDMC), Gasdermin D (GSDMD), Gasdermin E (GSDME, also called DFNA5), and pejvakin (PJVK, also called DFNB59) [3,52,53]. GSDM family members are expressed differently in different tissues and cells, and except for PJVK, all Gasdermins have conserved double domain arrangement: C-terminal domain and N-terminal domain, and the N-terminal domain has pore-forming activity [3,48,54,55]. The N-terminal domain of GSDMD is lipophilic and can bind with phosphatidylinositol phosphate, phosphatidylinoseroic acid, and cardiolipin, which makes the cell membrane form pores with the size of 10–14 nm, resulting in cell pyroptosis via the release of cell contents [8,56,57,58]. 

While GSDM family members are expressed differently in different tissues and cells, GSDMD is widely expressed in the cytosol of various cells and tissues [59]. It is the most important protein in the whole GSDM family as the common substrate protein of mouse caspase-1/11 or human caspase-4/5 and -1 [60]. GSDMD has 487 amino acids with 53 kDa weight and consists of 30 kDa N-terminal domain and 22 kDa C-terminal domain. Normally, C-terminal of GSDMD is connected to the N-terminal domain by a long loop, leaving GSDMD in an inactive autosuppressive state [56,59,61,62], while full-length GSDMD can be cleaved into two separate domains by caspase-1/11 in pyroptosis [59]; thereinto, N-terminal directly executes forming of membrane pore to induce pyroptosis [8]. However, neither C-terminal domain nor the full length GSDMD causes pyroptosis [61,62]. Consequently, GSDMD is the only substrate of inflammatory caspases, and the cleavage of GSDMD is a reliable marker of pyroptosis mediated by inflammatory caspases and inflammasome activation [8,58,63,64]. Therefore, GSDMD has become an important target for the intervention of pyroptosis. 

#### 2.2.3. Canonical and Noncanonical Inflammasome

The canonical inflammasome is a kind of multiprotein complex that mediates innate immune response in mammals. Its most prominent function is to recruit and activate caspase-1, promote the maturation of IL-1β and IL-18, and then generate inflammatory response. The inflammasome consists of NOD-like receptors (NLRs), apoptosis-associated Speck-like protein containing a CARD (ASC), and caspase-1 [65,66,67,68]. There are four intracellular receptor proteins to assemble the inflammasome, including NOD-like receptor protein 1 (NLRP1), NOD-like receptor protein 3 (NLRP3), NOD-like receptor C4 (NLRC4), and absent in melanoma 2 (AIM2) inflammasome [69]. The receptor is composed of three homologous domains: the N-terminal pyrin domain (PYD), the central nucleotide-binding oligomerization domain, and the C-terminal leucine repeat (LRR) [70]. In the process of canonical inflammasome assembly, NLRP3 recruits ASC through interaction with homotypic PYD domain and induces ASC to aggregate into macromolecular spots [71]. Subsequently, the assembled ASC recruits pro-caspase-1 through homotypic card domain interaction to form NLRP3-ASC-caspase-1 protein complex [72] that is known as NLRP3 inflammasome consisted of “sensor”, “adaptor”, and “effector” (Figure 2), respectively [73]. 

In the canonical pathway of pyroptosis, when cytosolic pathogen recognition receptors (PRRs), NLRP1b, NLRP3, NLRC4, AIM2, or Pyrin are stimulated by the corresponding PAMPs and DAMPs, these proteins recruit ASC and pro-caspase-1 to assemble into inflammasomes. The NLRP3 inflammasome is described in detail, and its assembly and activation process is divided into two steps. Firstly, pathogen-associated molecular patterns (PAMPs) or damage-associated molecular patterns (DAMPs) are recognized by Toll-like receptor 4 (TLR4) and activate the nuclear factor kappa-B (NF-κB) pathway, leading to increased transcription of NLRP3, pro-caspase-1, pro-IL-1β, and pro-IL-18 [74]. Secondly, under the further stimulation of immune and inflammatory molecules, the NLRP3 protein is oligomerized and assembled with ASC and pro-caspase-1 to form the NLRP3 inflammasome [75]. Formation of NLRP3 inflammasome results in cleavage of pro-caspase-1 to form the active form of caspase-1 and promotes cleavage of pro-IL-1β and pro-IL-18 to form IL-1β and IL-18 mature bodies, leading to a cascade of immune or inflammatory responses [76]. 

In the noncanonical pathway of pyroptosis, the complex of LPS-pro-caspase-11 was recognized as a noncanonical inflammasome, and LPS directly activates caspase-11, which is independent to TLR4 signaling pathway [77]. When caspase-11 mediates noncanonical pyroptosis, it leads to cleavage of GSDMD, and GSDMD-N executes pyroptosis to induce the release of cell contents. However, noncanonical pyroptosis still requires the help of the NLRP3 inflammasome that activates caspase-1 to induce maturation of pro-IL-1β and pro-IL-18 [77].

### 2.3. Mechanism of Pyroptosis

#### 2.3.1. Canonical Pyroptosis Pathway

Canonical pyroptosis is mediated by caspase-1, mainly in macrophages, and its key steps are the recruitment and activation of caspase-1 (Figure 3). Taking NLRP3 inflammasome as an example, when NLRP3 protein is stimulated by specific PAMPs and DAMPs, NLRP3 protein recruits ASC and pro-caspase-1 and assembles into NLRP3 inflammasome with the assistance of NIMA-related kinase 7 (NEK7) [78]. NLRP3 inflammasome assembly activates pro-caspase-1, and the activated caspase-1 can not only mediate the maturation and secretion of IL-1β and IL-18, but also directly cleave GSDMD to produce GSDMD-N [79]. Subsequently, GSDMD-N binds to phosphatidylinositol, phosphatidic acid, and phosphatidylserine on the inner surface of the membrane through membrane lipid interaction and forms oligomeric pores (GSDMS pore) with an inner diameter of 10~20 nm in the lipid bilayer. Then, LDH, IL-1β, and IL-18 are leaked out through the pores as well as other small cytosolic proteins and eventually cause pyroptosis [57]. Meanwhile, a large number of holes in the plasma membrane lead to the connection between the inner and outer membrane, forming a non-selective membrane channel, resulting in the efflux of K^+^ ions, which imbalance the ion concentration on both sides of the plasma membrane and cause a large amount of water to enter the cell, causing cell swelling and the eventual death of cells [8,80]. Under normal conditions, K^+^ efflux is generally considered to be both sufficient and necessary for NLRP3 inflammasome activation [81]. In addition, NLRC4, which is recognized by DAMPs or PAMPs, can directly activate caspase-1 to promote canonical pyroptosis [82]. 

#### 2.3.2. Noncanonical Pyroptosis Pathway

Numerous studies have found that caspase-1-independent pyroptosis pathway also exists in cells, which is named noncanonical pyroptosis and mediated by direct activation of caspase-4/5/11 under the action of LPS [83]. When Gram-negative bacteria infects mice, LPS is transferred by vesicles and enters the infected cells [84]. Mouse caspase-11 has CARD that can directly recognize the lipid A of LPS in the cytoplasm. After specific binding, caspase-11 oligomerized and activated, thereby mediating pyroptosis [48]. The functions of human caspase-4/5 and mouse caspase-11 are the same in mediating noncanonical pyroptosis [85]. Thus, caspase-4/5 also directly binds to LPS and promotes its own oligomerization and activation [86]. The activated caspase-4/5/11 can act on GSDMD and generate GSDMD-N fragments, thereby leading to cell membrane perforation and inducing pyroptosis [48,84,87,88,89]. In the pathway of canonical pyroptosis, caspase-1 cleaves IL-1β and IL-18 precursors to form active IL-1β and IL-18 and also cleaves GSDMD to produce N terminal; meanwhile, in the pathway of noncanonical pyroptosis, GSDMD cleavage is completed by the activated caspase-11 (Figure 4), but the cleavage of IL-1β and IL-18 precursor remains to depend on caspase-1 [43]. Thus, the noncanonical pyroptosis pathway depends on the activation of caspase-4/5/11 that lacks ASC participation, which is different from the canonical pathway. For human infection, activated caspase-4/5/11 can open Pannexin-1 channels to induce K^+^ efflux, which leads to NLRP3 inflammasome activation, promotes IL-1β and IL-18 maturation, and produces GSDMD-N to execute pyroptosis [48,79,83,90,91,92,93]. Meanwhile, ATP released from Pannexin-1 channels can activate P2X7R and promote K^+^ efflux, which in turn further promotes inflammasome assembly and triggers pyroptosis [78,94]. In addition, GSDMD-N can indirectly activate caspase-1 through the NLRP3-ASC-csapase-1 pathway, thereby promoting the maturation of IL-1β and IL-18 [95]. Therefore, new studies increasingly discover that caspase-4/5/11 is both a receptor and an effector molecule in the pathway of noncanonical pyroptosis [48]. 

## 3. Inhibitors of Pyroptosis

### 3.1. Chemical Agents as Inhibitors of Pyroptosis

Small molecule drugs mainly refer to chemically synthesized drugs, usually organic compounds with a molecular weight of less than 1000. They are usually signal transduction inhibitors, which can specifically block signal transduction pathways, and their chemical properties determined that small-molecule drugs have good therapeutic properties and pharmacokinetic properties [96]. In recent years, the research on small-molecule drugs to inhibit pyroptosis has received more and more attention.

NLRP3 inflammasome is a key upstream pathway of pyroptosis, and NLRP3 inhibitors provide a potential therapeutic approach for NLRP3-driven diseases by alleviating pyroptosis [97,98,99,100,101]. The literature has reported some NLRP3 inhibitors as shown in Table 1. CFTR(inh)-172 (C172) is an inhibitor for the cystic fibrosis transmembrane conductance regulator (CFTR) channel, which can block NLRP3 activation in a CFTR-independent manner [98]. The CY-09 can directly bind to NLRP3 to inhibit its assembly and activation and can bind to Walker A site in the NACHT domain to inhibit ATPase activity [98]. OLT1177, a β-sulfonyl nitrile compound, specifically inhibits the oligomerization and activation of typical and atypical NLRP3 inflammasomes in vitro and alleviated LPS-induced systemic inflammation in vivo [100,102]. Some sulfonylurea-containing compounds, such as glibenclamide, block IL-1β processing in response to prototypic NLRP3 stimuli, such as LPS and ATP, and block IL-1β release from human monocytes [103,104]. A compound was developed by Pfizer, called CRID3 (also known as CP-456,773) and renamed MCC950. MCC950 effectively inhibits the release of inflammatory factors IL-18 and IL-1β by inhibiting NLRP3 activation and blocks ASC oligomerization in vitro [100,105]. Other sulfonylureas, such as glyburide, sulofenur, and glimepiride, also inhibit NLRP3 signaling [106,107]. Dapagliflozin is an SGLT2 inhibitor that inhibits the NLRP3/ASC pathway and therefore may inhibit pyroptosis [108]. INF4E, a newly synthesized small-molecule inhibitor of NLRP3 inflammasome, significantly reduces myocardial infarct size and LDH release and attenuates the formation of NLRP3 inflammasome in a time-dependent manner [109]. INF39 is a covalent inhibitor that attenuates NLRP3 structural changes detected by BRET and inhibits NLRP3 ATPase activity, whereas HS-203873 attenuates NLRP3 ATPase activity and signaling [110,111,112]. Tranilast is an anti-allergic drug that appears to interact with the NLRP3 NACH T domain to disrupt NLRP3 intermolecular interactions and block NLRP3 oligomerization [113]. RRx-001 was developed as an anti-cancer molecule that can covalently interact with NLRP3 and block the NLRP3–NEK7 interaction [114]. The hydroxyl sulfonamide JC-171 down-regulates the expression of NLRP3-dependent IL-1β and reduces the level of ASC that is lowered by NLRP3, thereby destroying the NLRP3-ASC PPI [115]. The vinyl sulfone Bay 11-7082, the sesquiterpene lactone parthenolide, and the benzoxathiole derivative BOT-4-one all block the ATPase activity of NLRP3 [116,117]. 3, 4-methylenedioxy-β-nitrostyrene (MNS) is a Syk kinase inhibitor but inhibits NLRP3 activity independently of Syk by directly interacting with NLRP3 to block ATPase activity [118]. 

Caspase-1 is an important mediator of the classical pyroptosis pathway and is involved in the development of a variety of inflammatory diseases in vivo. As shown in Table 2, caspase-1 inhibitors were recorded in some reports. 1-(S)-(S)-2-2-3, 3-dimethyl-butanoyl)-pyrrolidine-2-carboxylic acid ((2R,3S)-2-ethoxy-5-oxo-tetrahydro-furan-3-yl)-amide (VX-765) was an orally absorbed prodrug of (S)-3-(-3,3-dimethyl-butanoyl)-pyrrolidin-2yl]-methanoyl}-amino)-4-oxo-butyric acid (VRT-043198), and VRT-043198 is a potent and selective inhibitor of the interleukin-converting enzyme/caspase subfamily [119]. The specific caspase-1 inhibitor VX-765 reduced ox-LDL-mediated pyroptosis of VSMCs [120]. Boc-D-FMK inhibits caspase-1 activity, reduces mitochondrial dysfunction, and inhibits the production of downstream proinflammatory cytokines [97]. Ac-YVAD-cmk is a selective and irreversible inhibitor of caspase-1 that prevents caspase-1 activation [121]. In addition, as mentioned above, parthenolid also has the function of inhibiting caspase-1 [116].

GSDMD is a key protein in pyroptosis. Screening and designing small molecule inhibitors specifically targeting GSDMD can prevent pyroptosis. As shown in Table 3, these are some GSDMD inhibitors described in literature. Necrosulfonamide (NSA) directly binds to GSDMD and inhibits the oligomerization of GSDMD-N, reducing the opening degree of cell membrane pores and the release of downstream inflammatory factors, but does not affect the expression of upstream NLRP3 and caspase-1 and the cleavage of GSDMD protein [122]. LDC7559 is a small molecule compound selected from the search for inhibitors of the special death form of human neutrophils, which blocks the toxicity of GSDMD-N and reduces inflammation [123]. Disulfiram, a drug used to treat alcohol addiction, can effectively inhibit the formation of GSDMD pores in human and mouse cells, thereby inhibiting pyroptosis [124]. It has been reported that fumarate, a tricarboxylic acid cycle intermediate, can acylate cysteine in GSDMD and prevent its interaction with cysteine proteases and subsequent processing activation, thus inhibiting the occurrence of pyroptosis [125]. Bay 11-7082 was previously identified as an NF-κB inhibitor, which can directly lead to covalent modification of the cysteine 191/192 residue of GSDMD, interfere with the formation of GSDMD pore and IL-1β secretion, and effectively inhibit pyroptosis [126].

### 3.2. Natural Product as Inhibitor of Pyroptosis

Natural products are metabolites produced by organisms in nature and have been an important source of new drug discovery. For the development of new drugs, research has reported some natural products as inhibitors of pyroptosis (Table 4). Baicalin, a flavonoid isolated from the rhizome of Scutellariae radix, can inhibit NLRP3 inflammasome activation and pyroptosis by affecting the activity of PKA in macrophages [127]. Dihydromyricetin is a natural flavonoid isolated from Ampelopsis grossedentata and inhibits pyroptosis by increasing cell viability, reducing LDH and IL-1β release, protecting cell membrane integrity, and eliminating caspase-1 cleavage and subsequent IL-1β maturation [128]. Punicalagin is the main component of pomegranate polyphenols, which can specifically prevent pyroptotic membrane permeability and may therefore interfere with the insertion or oligomerization of GSDMD-N in the plasma membrane [129]. Punicalagin is also an effective antioxidant, and reactive oxygen species (ROS) promote pore assembly through GSDMD-N [130]. Thus, punicalagin can also affect pyroptosis by scavenging ROS. Moreover, punicalagin could down-regulate the expression of NOX4 and inhibit TXNIP/NLRP3 pathway-mediated pyroptosis [131]. Catalpol, an iridoid glycoside rich in the root of rehmanniae radix, can inhibit oxidative stress, inflammation, and pyroptosis through AMP-activated protein kinase(AMPK)/SIRT1/NF-κB pathway [132]. Geniposide is one of the active ingredients extracted from the dried and mature fruit of Gardenia jasminoides that can effectively block oxidative stress and inflammatory response accompanied by pyroptosis through inhibiting the APMK/SIRT1/NF-κB pathway [133]. The total flavones of Abelmoschus maniho from Abelmoschus manihot can inhibit pyroptosis by regulating METTL3-dependent m6A modification, NLRP3-inflammasome activation and PTEN/PI3K/Akt signaling pathway [134]. Oridonin is a bioactive natural compound isolated from Rabdosia rubescens, which can directly interact with NLRP3, covalently modify NLRP3 residue C279 in the NACHT domain, and attenuate the interaction between NLRP3 and Never to inhibit the formation of inflammasomes [135].

Aesculin is the main component of ash bark, which promotes the phosphorylation of protein kinase B (Akt)/glycogen synthase kinase 3β (GSK3β), inhibits the phosphorylation of nuclear factor-κb (NF-κB), and significantly reduces the expression levels of NLRP3, caspase-1, GSDMD, and IL-1β, thereby inhibiting pyroptosis and alleviation of inflammatory symptoms [136]. Notoginsenoside R1 is the main component of panax notoginseng, which can inhibit the NF-κB signaling pathway, block the activation of NLRP3 inflammasome, prevent the cleavage of caspase-1 and IL-1β, reduce the expression of the N-terminal domain of GSDMDN, and improve the cell function of rat nucleus pulposus cells, thereby inhibiting pyroptosis [137]. Ginsenoside Rb1 is another important component of Panax notoginseng, which can regulate the nuclear transcription factor E2-related factor 2 (Nrf2)/antioxidant response element signaling pathway to reduce the pyroptosis by inhibiting calcium overload [138]. Quercetin is a natural flavonoid widely distributed in vegetables and fruits, which can inhibit LPS/ATP-induced NLRP3 inflammasome activation and reduce the cleavage of GSDMD protein and the secretion of IL-1β and IL-18 [139]. Resveratrol is a polyphenolic compound that occurs naturally in a variety of plants, especially red grape skins, which can inhibit the activation of NLRP3 inflammasome and down-regulate the expression of caspase-1, thereby reducing the cleavage of GSDMD protein and the release of IL-1β and IL-18, and finally significantly inhibiting LPS/ATP-induced pyroptosis of macrophages [140]. Alliin is an organic sulfur compound extracted from garlic, which can inhibit NLRP3 inflammasome activation and reduce the secretion of IL-1β and IL-18 in mouse macrophages induced by LPS [141]. Curcumin is a natural polyphenol extracted from the rhizome of Curcuma longa, which can inhibit the NLRP3 inflammasome-mediated pyroptosis by regulating TLR4/NF-κB signaling pathway in mouse microglia [142]. Luteolin is a natural flavonoid found in a variety of plants, which can inhibit NLRP3 inflammasome activation and reduce the expression of the N-terminal domain of GSDMD and IL-1β by inhibiting Nrf2 and NF-κB signaling pathways [143]. Salidroside is extracted from Rhodiola rosea, which can reduce the levels of IL-1β, IL-18, and GSDMD. It can inhibit the NLRP3/caspase-1/GSDMD axis-mediated pyroptosis [144]. Salvianolic acid is extracted from salviae miltiorrhizae, which can reduce the cleavage of GSDMD-FL to GSDMD-N and inhibit the NLRP3 inflammasome [145]. Oroxylin is extracted from oroxylum indicum, which can reduce the activation of NLRP3 inflammasome and caspase-1 and inhibit GSDMD-mediated pyroptosis and the release of IL-1β and IL-18 [146]. Glycyrrhizin is a triterpene compound isolated from licorice, which can inhibit pyroptosis by inhibiting caspase-1/GSDMD signaling pathway [147]. In addition, many natural plant components with antioxidant and anti-inflammatory effects can inhibit pyroptosis by inhibiting the NLRP3/GSDMD signaling pathway, including paeoniflorin extracted from paeonia lactiflora pall., berberine extracted from coptis chinensis, lycorine isolated from lycoris radiata, isoliquiritigenin isolated from licorice, and protocatechuic acid present in many vegetables and fruits [148,149,150,151,152].

### 3.3. Herb Medicine as Inhibitor of Pyroptosis

Herbal medicine, especially traditional Chinese medicine, has become an important repository for new drug discovery. Some representative formulas of TCM have been found to play a role in pyroptosis regulation as shown in Table 5. For example, Baihu Guizhi Tang inhibits the activation of NLRP3 inflammasome and suppresses cell pyroptosis [153]. Daphnes Cortex inhibits macrophage pyroptosis by regulating TLR4/NF-κB/NLRP3 signaling pathway [154]. Kuijieling Tang reduces the mRNA levels of Caspase-1, ASC, IL-1β, and IL-18 and significantly reduces the protein expression of NLRP3, caspase-1, GSDMD-N, IL-1β, and IL-18 [155]. Shenling Baizhu San inhibits the production of pro-inflammatory factors IL-1β, IL-18, and TNF-a and reduces the mRNA expression levels of NLRP3, ASC, and GSDMD-N [156]. Yushi Anchang Fang can down-regulate the expressions of NLRP3, ASC, and caspase-1 to close to normal levels, thereby inhibiting pyroptosis [157]. Due to the flexible composition of TCM prescriptions, we just listed the above as some representative prescriptions studied in recent years. As shown in Table 5, TCM formulas often consist of two or more herbs, and effective components are also complex. However, the complexity of effective components also makes the characteristics of multi-target inhibition of pyroptosis of TCM formulas, which reflects the pan-inhibitory effect on pyroptosis. Therefore, we suggest that TCM should be considered as a candidate library for development of pyroptosis blockers.

## 4. Conclusions and Future Perspectives

Pyroptosis is a newly discovered form of programmed death of inflammatory cells. The canonical pyroptosis pathway of pyroptosis is mediated by caspase-1, and the noncanonical pyroptosis pathway is mediated by caspase-4/5 and caspase-11. Pyroptosis is characterized by the activation of various caspases through inflammasomes, resulting in the cleavage and multimerization of various Gasdermin family members to perforate the cell membrane, which in turn causes cell death. It is different from apoptosis in that pyroptosis occurs more rapidly and is accompanied by the release of a large number of proinflammatory cytokines. At the same time, inflammatory cytokines recruit other inflammatory cells and expand the inflammatory response. Actually, pyroptosis is an innate immune response that fights against infection and plays an important role in the occurrence and development of the normal function of the immune system in the body. Therefore, the regulation of pyroptosis to keep it within a reasonable range is helpful to the treatment of diseases and the health of the body. However, excessive pyroptosis occurs and leads to immune disorders in many pathological conditions. It is crucial to follow the mechanisms of pyroptosis and understand the potential drugs that inhibit excessive pyroptosis and their mechanisms, so as to keep pyroptosis at a moderate level. Since pyroptosis is an active and orderly form of cell death regulated by intracellular signals, its program can be interfered with or reprogrammed, which provides a theoretical basis for regulating pyroptosis and, therefore, makes it possible to study drugs that regulate or block pyroptosis. While some small molecule compounds have been reported that inhibit pyroptosis, there is still a long time before clinical drugs are developed. By contrast, some natural products and herbal medicines have significant inhibitory effects on pyroptosis, which may become a new growth point in the development of pyroptosis inhibitory drugs, and most of them have been used clinically for a long time. Therefore, more attention should be paid to herbal medicines for research and development of pyroptosis blockers.

## Figures and Tables

**Figure 1 ijms-23-16115-f001:**
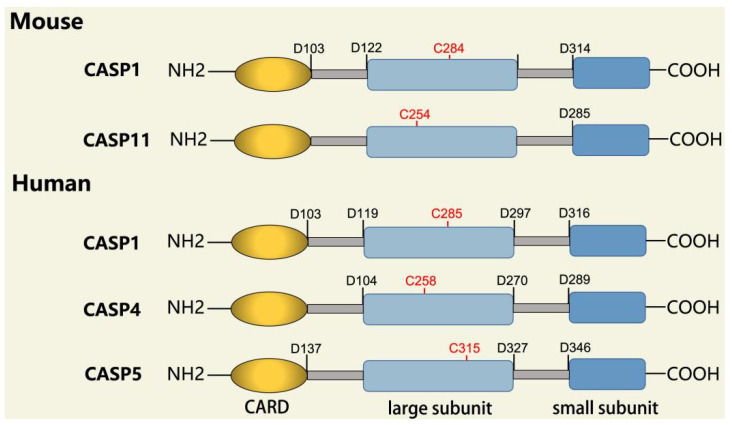
Domain structure of mouse (caspases-1, -11) and human (caspases-1, -4, -5) inflammatory caspases. A CDL connects N-terminal CARD to the protease domain that is composed of a large subunit (p20) and a small subunit (p10) separated by an IDL. Each caspase contains some autocleavage sites (black site) within the linker sequences. The catalytic cysteine (red site) is located within the large protease subunit, while the dimerization interface is within the small subunit.

**Figure 2 ijms-23-16115-f002:**
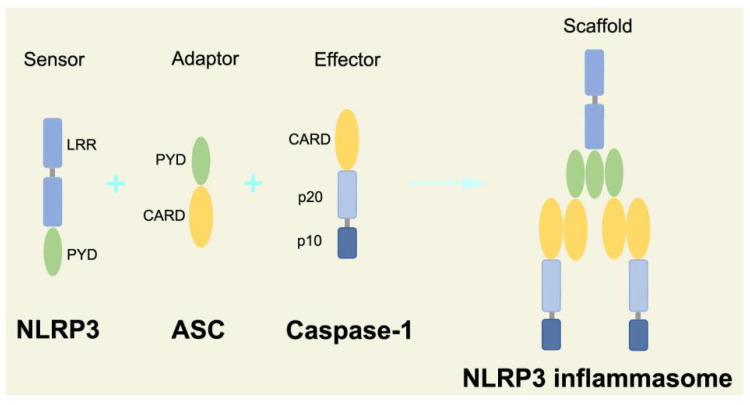
Structure and assembly of NLRP3 inflammasome. NLRP3 inflammasome consists of NLRP3, ASC, and caspase-1, which are known as sensor, adaptor, and effector, respectively.

**Figure 3 ijms-23-16115-f003:**
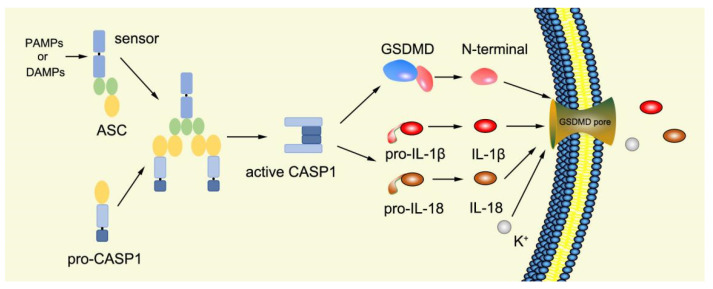
Mechanism of canonical pyroptosis pathway.

**Figure 4 ijms-23-16115-f004:**
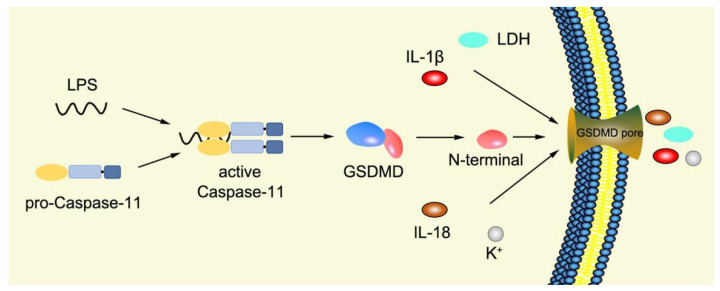
Mechanism of noncanonical pyroptosis pathway.

**Table 1 ijms-23-16115-t001:** Pyroptotic inhibitors targeted to NLRP3.

Drug	Structure	Mechanism of Action	IC_50_	References
C172	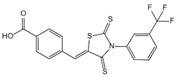	Blocks NLRP3 activation in a CFTR-independent manner	0.3 μM	[98]
CY-09	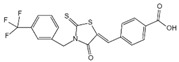	Binds to NLRP3 to inhibit its assembly and activation, and binds to Walker A site in the NACHT domain to inhibit caspase activity	6 μM	[98]
OLT1177	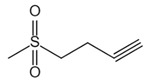	Inhibits the oligomerization and activation of typical and atypical NLRP3 inflammasomes, and alleviates LPS-induced systemic inflammation	/	[100,102]
MCC950	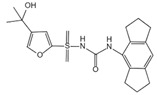	Inhibits the release of IL-18 and IL-1β by inhibiting NLRP3 activation, and blocks ASC oligomerization	7.5 × 10^−3^ μM	[98,104]
Glyburide	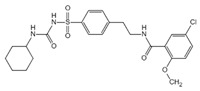	Inhibits NLRP3 signaling	20 µM	[107]
sulofenur	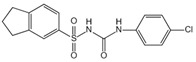	Inhibits NLRP4 signaling	0.034 μM	[106]
Glimepiride	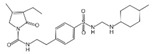	Inhibits NLRP5 signaling	5.4 × 10^−3^ μM	[106]
Dapagliflozin	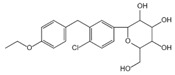	Inhibits the NLRP3/ASC pathway	1.6 × 10^−3^ μM	[108]
INF4E	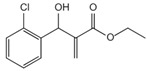	Reduces myocardial infarct size and LDH release, and attenuates the formation of NLRP3 inflammasome in a time-dependent manner	/	[109]
INF39	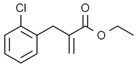	Attenuates NLRP3 structural changes detected by BRET and inhibits NLRP3 ATPase activity	10 μM	[110,111,112]
HS-203873	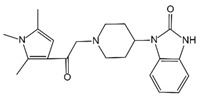	Attenuates NLRP3 activity and signal transduction	50 μM	[110,111,112]
Tranilast	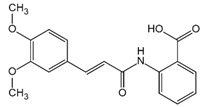	Appears to interact with the NLRP3 NACHT domain to disrupt NLRP3 intermolecular interactions and block NLRP3 oligomerization	25 μM	[113]
RRx-001	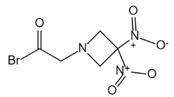	Covalently interacts with NLRP3 and blocks the NLRP3-NEK7 interaction	0.117 μM	[114]
JC-171	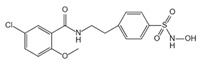	Down-regulates the expression of NLRP3-dependent IL-1β and reduces the level of ASC that is lowered by NLRP3	8.45 μM	[115]
Bay 11-7082	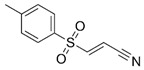	Blocks the ATPase activity of NLRP3	3 μM	[116]
Parthenolide	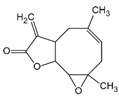	Blocks the ATPase activity of NLRP3	1.4 μM	[117]
BOT-4-one	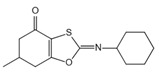	Blocks the ATPase activity of NLRP3	1.28 μM	[117]
MNS	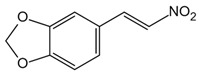	Inhibits NLRP3 activity independently of Syk by directly interacting with NLRP3 to block caspase activity	3 μM	[118]

**Table 2 ijms-23-16115-t002:** Pyroptotic inhibitors targeted to caspase-1.

Drug	Structure	Mechanism of Action	IC_50_	References
VX-765	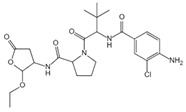	VX-765 is the prerequisite drug of VRT-043198, and it can specifically inhibit caspase-1 after in vivo conversion to VRT-043198	0.7 μM	[119,120]
Boc-D-FMK	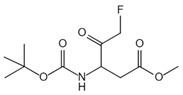	Inhibits caspase-1 activity	39 μM	[97]
Ac-yvad-cmk	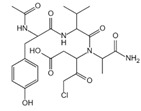	Prevents caspase-1 activation	/	[121]

**Table 3 ijms-23-16115-t003:** Pyroptotic inhibitors targeted to GSDMD.

Drug	Structure	Mechanism of Action	IC_50_	References
NSA	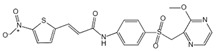	Binds to GSDMD and inhibits the oligomerization of GSDMD-N	/	[122]
LDC7559	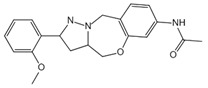	Blocks the toxicity of GSDMD-N	/	[123]
Disulfiram	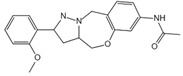	Inhibits the formation of GSDMD pores	<1 M	[124]
Fumarate	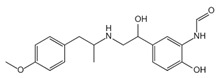	Acylate cysteine in GSDMD and prevents its interaction with cysteine proteases and subsequent activation	0.003 μM	[125]
Bay11-7082	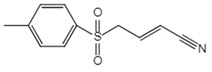	Covalent modification of the cysteine 191/192 residue of GSDMD, interferes with the formation of GSDMD pore and IL-1β secretion	10 μM	[126]

**Table 4 ijms-23-16115-t004:** Natural products as inhibitors of pyroptosis.

Drug	Structure	Mechanism of Action	Source	References
Baicalin	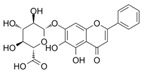	Reduces LDH and IL-1β release, eliminate caspase-1 cleavage and maturation	Scutellariae radix	[127]
Dihydromyricetin	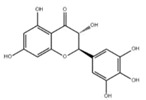	Reduces LDH and IL-1β release, eliminate caspase-1 cleavage and IL-1β maturation	Ampelopsis grossedentata	[128]
Punicalagin	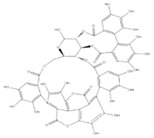	Prevents pyroptotic membrane permeability and may interfere with the insertion or oligomerization of GSDMD-N in the plasma membrane	Pomegranate	[129,130,131]
Catalpol	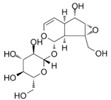	Inhibits AMPK/SIRT1/NF-κB pathway	Rehmanniae radix	[132]
Geniposide	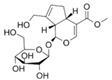	Inhibits AMPK/SIRT1/NF-κB pathway	Gardenia jasminoides	[133]
Total flavones of Abelmoschus manihot	N/A	Inhibits NLRP3-inflammasome activation and PTEN/PI3K/Akt signaling pathway	Abelmoschus manihot	[134]
Oridonin	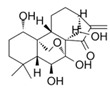	Interacts with NLRP3, covalently modifies NLRP3, and attenuates NLRP3 activation	Rabdosia rubescens	[135]
Aesculin	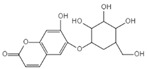	Promotes the phosphorylation of Akt/GSK3β, inhibits the phosphorylation of NF-κB, and significantly reduces the expression levels of NLRP3, caspase-1, GSDMD, and IL-1β	Ash bark	[136]
Notoginsenoside R1	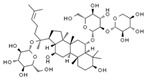	Inhibits the NF-κB signaling pathway, blocks the activation of NLRP3 inflammasome, prevents the cleavage of caspase-1 and IL-1β, reduces the expression of GSDMDN	Panax notoginseng	[137]
Ginsenoside Rb1	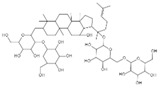	Regulates Nrf2 /antioxidant response element signaling pathway and reduces the pyroptosis	Panax notoginseng	[138]
Quercetin	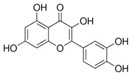	Inhibits LPS/ATP-induced NLRP3 inflammasome activation and suppresses the expression of GSDMD, NRPS, and IL-1β	Vegetables and fruits	[139]
Resveratrol	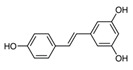	Inhibits the expression of Caspase-1, reduces the cleavage of GSDMD protein and the release of IL-1β and IL-18; suppresses the expression of p62 and the activation of NLRP3 inflammasome	Red grape skins	[140]
Alliin	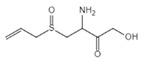	Inhibits NLRP3 inflammasome activation and reduces the secretion of IL-1β and IL-18	Garlic	[141]
Curcumin	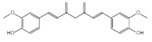	Inhibits the NLRP3 inflammasome-mediated pyroptosis and reduces TLR4/NF-κB pathway	Curcuma longa	[142]
Luteolin	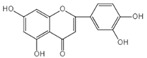	Inhibits NLRP3 inflammasome activation and reduces the expression of the N-terminal domain of GSDMD and IL-1β by inhibiting Nrf2 and NF-κB signaling pathways	A variety of plants	[143]
Salidroside	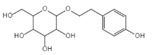	Reduces the levels of IL-1β, IL-18, and GSDMD, and inhibits the NLRP3/caspase-1/GSDMD axis mediated pyroptosis	Rhodiola rosea	[144]
Salvianolic acids	N/A	Reduces the cleavage of GSDMD-FL to GSDMD-N and inhibits the NLRP3 inflammasome/pyroptosis activation	Salviae miltiorrhizae	[145]
Oroxylin	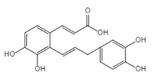	Reduces the activation of NLRP3 inflammasome and caspase-1, and inhibits GSDMD-mediated pyroptosis and the release of IL-1β and IL-18	Oroxylum indicum	[146]
Glycyrrhizin	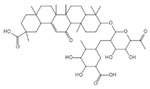	Inhibits caspase-1/GSDMD signaling pathway	Licorice	[147]
Paeoniflorin	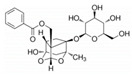	Inhibits the NLRP3/GSDMD signaling pathway	Paeonia lactiflora pall	[148]
Berberine	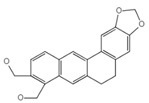	Inhibits the NLRP3/GSDMD signaling pathway	Coptis chinensis	[149]
Lycorine	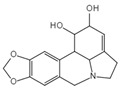	Inhibits the NLRP3/GSDMD signaling pathway	Lycoris radiata	[150]
Isoliquiritigenin	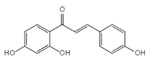	Inhibits the NLRP3/GSDMD signaling pathway	Licorice	[151]
Protocatechuic Acid	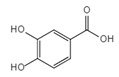	Inhibits the NLRP3/GSDMD signaling pathway	Vegetables and fruits	[152]

**Table 5 ijms-23-16115-t005:** TCM formulas as inhibitors of pyroptosis.

Formulas	Consists of Herbs	Mechanism of Action	References
Baihu Guizhi Tang	Anemarrhena asphodeloides, Gypsum, Licorice, Rice, Cinnamon twig	Inhibits the activation of NLRP3 inflammasome	[153]
Daphnes Cortex	Stem and root bark of Daphne giraldii Nitsche	Regulates TLR4/NF-κB/NLRP3 signaling pathway	[154]
Kuijieling Tang	Ilicis Rotundae Cortex, Rhizoma Atractylodis, Macrocephalae, Paeoniae Radix Alba, Hirudo, and Radix Glycyrrhizae Preparata	Reduces the expression of NLRP3, caspase-1, GSDMD-N, IL-1β and IL-18	[155]
Shenling Baizhu San	Semen lablab album, Atractylodes macrocephala, poria cocos, Licorice, Radix platycodi, Lotus seed, Ginseng, Fructus amomi, Yam, Coix seed	Inhibits IL-1β, IL-18, and TNF-α; and reduces the mRNA expression levels of NLRP3, ASC, and GSDMD-N	[156]
Yushi Anchang Fang	Scutellaria baicalensis, Astragalus, Atractylodes, Licorice, Radix Paeoniae Alba, Cinnamon, Pueraria, Carbonized Catnip, Radix Sanguisorbae, Bletilla Striata, Frankincense, Herba Patriniae, Sargent Gloryvine, and Panax Notoginseng powder	Down-regulates the expressions of NLRP3, ASC, and caspase-1	[157]

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
