# Peer review of "Molecular Characteristics of Cell Pyroptosis and Its Inhibitors: A Review of Activation, Regulation, and Inhibitors"

_ijms, 2022, doi:10.3390/ijms232416115_

Round 1
Reviewer 1 Report
This is an interesting review on the molecular aspects of pyroptosis and its potential inhibitors.
I suggest to the authors to add figures indicating and explaining the key molecules participating in canonical and non-canonical pathways of pyroptosis in order to improve their work and make it more comprehensive to the readers. Also, I suggest to add extra columns in tables describing the type of study that the inhibitor was tested and any potential clinical significance if there is such evidence available.
Author Response
We attach great importance to the reviewer's valuable advice, and we have made corresponding changes to it. For the first suggestion, we added structure diagram of inflammatory caspases and pathway diagram of pyroptosis. As for the second opinion, because the relevant literature did not mention it, we are not going to make any additional revisions.
Reviewer 2 Report
Wei et al have complied an extensive review of present literature in the present review article entitled as " Molecular characteristics of cell pyroptosis and its inhibitors: a review of activation, regulation and inhibitors.
Pyroptosis remains a critically important phenomenon in regulation of innate immune response. The molecular signal transduction pathways involved in regulation of pyroptosis ultimately control the fate of cell death. Several small molecules targeting pyroptosis are under development. Authors have elegantly explained the canonical and non-canonical pathways of apoptosis. Most importantly, authors highlighted the potential use of natural products as inhibitors of pyroptosis. This review summarises the importance of further development of selective pyroptosis inhibitors from the lead natural products. I think this review is interesting and useful for broad audience.
Major observations:
1) Line 227, The definition of small molecule drugs under the section "chemical agents as inhibitors of pyroptosis appears technically incorrect. It should be rephrased for the sake of clarity and understanding for broader audience.
2) Line 198, Under mechanism of pyroptosis, authors discuss about "non-selective membrane channels that imbalance the ion concentration on both sides of the plasma membrane".. More insightful information is expected here explaining the role of leaky ion channels that alter the osmotic properties of the cell affecting cell health and inducing pathogenesis.
3) Table 1 and 2, The visual quality of structures is so poor and indistinct. It should be upgraded to high resolution and consistent throughout.
4) In table 1-2, the IC50 values are inconsistently represented. Font and size varies a lot. It should be fixed.
Minor observations:
1) Page 9, para 3.2, the botanical names of medicinal plants should be represented as per their convention.
2) Typo error should be corrected in Table 4, Mechanism of action of Punicalagin.
Author Response
Q1: Line 227, The definition of small molecule drugs under the section "chemical agents as inhibitors of pyroptosis appears technically incorrect. It should be rephrased for the sake of clarity and understanding for broader audience.
Answer: We attach great importance to the reviewer's valuable advice, and we have made corresponding changes to it. We amended the part of the description of small molecule drugs by adding "Small molecule drugs mainly refer to chemically synthesized drugs, usually organic compounds with a molecular weight of less than 1000, and they are usually signal transduction inhibitors.”.
Q2: Line 198, Under mechanism of pyroptosis, authors discuss about "non-selective membrane channels that imbalance the ion concentration on both sides of the plasma membrane". More insightful information is expected here explaining the role of leaky ion channels that alter the osmotic properties of the cell affecting cell health and inducing pathogenesis.
Answer: We have added an example of K+ efflux to explain the role of leaky ion channels in canonical pyroptosis pathway. The statement was amended from line 217 to line 223 in revised manuscript.
Q3: Table 1 and 2, The visual quality of structures is so poor and indistinct. It should be upgraded to high resolution and consistent throughout.
Answer: We have replaced the most structure charts with high quality pictures.
Q4: In table 1-2, the IC50 values are inconsistently represented. Font and size varies a lot. It should be fixed.
Answer: We have changed all the units of IC50 to µM.
Q5: Page 9, para 3.2, the botanical names of medicinal plants should be represented as per their convention.
Answer: We have amended these botanical names. Moreover, we added more examples of isolated phytochemicals engaged in pyroptosis in this section.
Q6: Typo error should be corrected in Table 4, Mechanism of action of Punicalagin.
Answer: Type errors were checked and corrected one by one carefully as well as mechanism of Punicalagin in Table 4.
Round 2
Reviewer 1 Report
The authors have adequately answered my concerns and made appropriate changes.